# Learning to Generate Human-Human-Object Interactions from Textual Descriptions

**Jeonghyeon Na,**[*] **Sangwon Baik,**[*] **Inhee Lee, Junyoung Lee, Hanbyul Joo**[†]

Seoul National University
[*]Equal Contribution   [†]Corresponding Author
{prom317,bsw1907,ininin0516,juncong,hbjoo}@snu.ac.kr

## Abstract

The way humans interact with each other, including interpersonal distances, spatial configuration, and motion, varies significantly across different situations. To enable machines to understand such complex, context-dependent behaviors, it is essential to model multiple people in relation to the surrounding scene context. In this paper, we present a novel research problem to model the correlations between two people engaged in a shared interaction involving an object. We refer to this formulation as Human-Human-Object Interactions (HHOIs). To overcome the lack of dedicated datasets for HHOIs, we present a newly captured HHOIs dataset and a method to synthesize HHOI data by leveraging image generative models. As an intermediary, we obtain individual human-object interaction (HOIs) and human-human interaction (HHIs) from the HHOIs, and with these data, we train an text-to-HOI and text-to-HHI model using score-based diffusion model. Finally, we present a unified generative framework that integrates the two individual model, capable of synthesizing complete HHOIs in a single advanced sampling process. Our method extends HHOI generation to multi-human settings, enabling interactions involving more than two individuals. Experimental results show that our method generates realistic HHOIs conditioned on textual descriptions, outperforming previous approaches that focus only on single-human HOIs. Furthermore, we introduce multi-human motion generation involving objects as an application of our framework.

## 1 Introduction

Human behavior in real-world environments is inherently social and context-dependent. People naturally interact with one another through structured patterns of interpersonal distance, spatial configuration, and motion, which are intuitively understood by humans but vary significantly across different situations. Understanding these nuanced, multi-human interactions is critical for AI systems that aim to interpret, simulate, or engage naturally in human-centric environments. While Human-Object Interactions (HOIs) [66, 57, 9, 32, 65, 63, 45, 26] and Human-Human Interactions (HHIs) [67, 23, 10, 35, 42, 10, 43, 54, 50, 36, 56] have been extensively studied in isolation, modeling interactions involving both multiple people and shared objects remains underexplored. In particular, dyadic interactions, where two individuals engage in a coordinated activity around a common object, are prevalent in everyday life, but have received relatively little attention. Examples include sitting together on a sofa, sharing an umbrella, or standing side by side at a whiteboard for discussion. In this paper, we present a novel research problem: modeling the coordinated behavior of two individuals interacting around a shared object. We refer to this formulation as Human-Human-Object Interactions (HHOIs). A core challenge in studying HHOIs is the absence of dedicated datasets. Existing HOI datasets [52, 15, 68, 16, 61] primarily feature single-human-object interactions, while

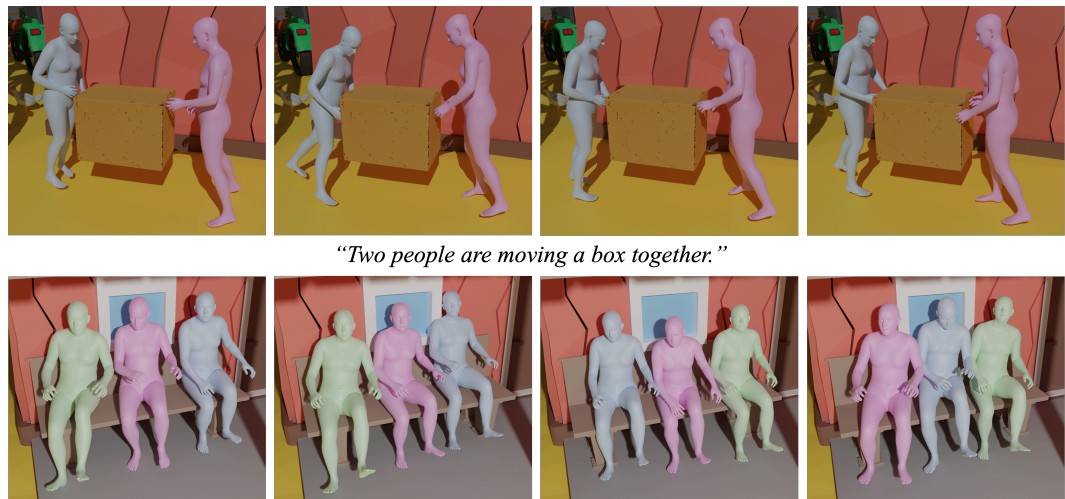

*"Two people are moving a box together."*

*"Three people are sitting on a bench"*

Figure 1: Results of our HHOI generation given object instances and text-prompt descriptions. Multiple humans in action are generated by jointly enforcing scene-level consistency across human-object interactions (HOIs) and human-human interactions (HHIs).

HHI datasets [30, 12, 14, 35, 55, 13, 38, 42] typically lack object context, often limited to dyadic conversational scenarios in standing poses.

To address the lack of diverse HHOI data, we introduce a newly collected 3D dataset captured using a multi-camera system, specifically designed to support the training and evaluation of HHOI models. In addition, we present a synthetic HHOI dataset generation pipeline that leverages pretrained image diffusion models [41] to complement real-world data, particularly for scenarios that are challenging to capture in studio environments. These diverse data sources are unified through a score-based diffusion model, enabling realistic generation of dyadic human-object interactions. Our model is conditioned on textual descriptions, enabling semantically grounded generation of HHOIs. Importantly, we further demonstrate that our framework can be extended beyond dyadic interactions to accommodate multi-human interactions, offering a scalable solution for modeling increasingly complex social behaviors. Experimental results show that our method produces more realistic and coherent interactions compared to existing baselines that model only single-human HOIs. As a potential application of our model, we present multi-human motion generation via Diffusion-Noise-Optimization [25], with our output HHOI as constraint.

Our contributions are summarized as follows: (1) Curated datasets for HHOI, along with methodologies for constructing them; (2) Score-based HHOI model that jointly captures both individuals' interactions with a shared object and their interaction with each other, conditioned on textual description; (3) Extension to multi-human HHOIs, enabling generation of interactions beyond dyadic settings; (4) Application to object interaction-aware multi-human motion generation.

## 2 Related Work

**Human-Object Interaction** Human-object interaction (HOI) aims to understand how humans interact with objects in the environment. This line of research is crucial for enabling machines to interpret and mimic human behaviors, thereby supporting the development of embodied AI agents and realistic digital human modeling with natural object manipulation. There have been considerable efforts to collect and scale up 3D HOI dataset, aiming to pursue a data-driven approach in this direction. Early work tries to capture the 3D HOI scenes in a controlled setup using marker [22], IMU [52, 15, 68, 16], multi-view capture system [18, 19, 61, 3], or hybridizing them [11, 28, 64, 63, 37]. In addition to real-world capture, synthetic datasets have also been introduced using game engines [6] and physics simulation [4]. More recently, automated pipelines have emerged to generate 3D HOI scenes from pre-trained 2D image models, significantly reducing capture effort and expanding scenario

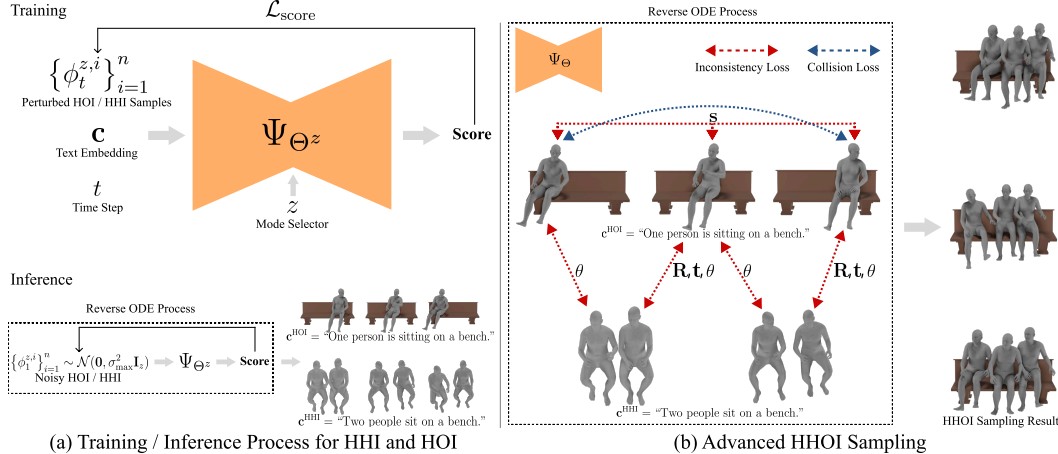

Training

$\mathcal{L}_{\text{score}}$

$\left\{ \phi_t^{z,i} \right\}_{i=1}^{n}$
Perturbed HOI / HHI Samples

$\mathbf{c}$
Text Embedding

$t$
Time Step

$\Psi_{\Theta^z}$

$z$
Mode Selector

Score

Inference

Reverse ODE Process

$\left\{ \phi_1^{z,i} \right\}_{i=1}^{n} \sim \mathcal{N}(\mathbf{0}, \sigma_{\max}^2 \mathbf{I}_s)$
Noisy HOI / HHI

$\Psi_{\Theta^z}$  Score

$c^{\text{HOI}}$ = "One person is sitting on a bench."

$c^{\text{HHI}}$ = "Two people sit on a bench."

(a) Training / Inference Process for HHI and HOI

Reverse ODE Process

$\Psi_\Theta$

Inconsistency Loss   Collision Loss

$\mathbf{S}$

$c^{\text{HOI}}$ = "One person is sitting on a bench."

$\theta$   $\mathbf{R}, \mathbf{t}, \theta$   $\theta$   $\mathbf{R}, \mathbf{t}, \theta$

$c^{\text{HHI}}$ = "Two people sit on a bench."

(b) Advanced HHOI Sampling

HHOI Sampling Results

Figure 2: **Method Overview.** (a) The training and inference process of the HOI/HHI part. (b) The advanced HHOI sampling process by introducing inconsistency loss and collision loss.

diversity [26, 17, 33, 27]. Various models [57, 9, 32, 65, 45, 58, 29, 60, 21] have been proposed to learn from the presented datasets, including models manipulating articulated objects [11, 28], and multiple objects at once [63, 28]. However, the majority of existing methods are limited to single-human interaction scenarios, while scenarios involving multiple humans with objects remain largely unexplored. Core4D [37] takes a step toward this direction by collecting HHOI data for collaborative tasks involving two people, but its scale and diversity are still limited.

**Human-Human Interaction** Modeling the interaction between individuals is essential to capture cooperative behaviors and social dynamics [24], which is crucial for developing embodied AI agents capable of natural human interaction. Previous works have primarily focused on dyadic human-human interactions, aiming either to reconstruct plausible interactions from images [12, 38], or to generate natural motion [30, 35, 55, 13, 42]. Recently, several methods have been proposed to extend the interaction modeling to more than two individuals [67, 23, 10, 43]. Many of these approaches are conditioned by contextual signals such as text [35, 42, 10, 43], music [44, 34], or predefined action-reaction roles [54, 50, 36, 56], yet they often struggle to capture interactions that are tightly coupled with a specific target object. MAMMOS [36] explores scene-conditioned multi-human motion generation, yet remains limited in scope and lacks complex interactions.

**Score-based Generative Models.** Score-based generative models [46, 47] estimate gradients of the data distribution for generative modeling, by introducing noise conditional score networks to learn score functions at multiple noise levels. Later work [48] generalized this approach using stochastic differential equations, providing a continuous-time formulation. This framework has proven highly effective for various generation tasks such as object rearrangement [53], object pose estimation [62], scene-graph generation [49] and human pose estimation [8]. Recent extensions apply score-based models to interactive settings, such as human-object interaction [27], and object-object interaction [1].

# 3 Method

Our HHOI model is structured as a combination of HOI model and HHI model. We first introduce how each component—HOI model and HHI model—is independently represented (Sec. 3.1). Specifically, we model HOI and HHI using score-based diffusion models [46, 48]. We then describe the training and inference procedures for each component (Sec. 3.2). Finally, we present a guided sampling method that integrates each component with inconsistency and collision constraints during the inference process to generate coherent and plausible HHOI configurations (Sec. 3.3).

## 3.1 HHOI Formulation

**Modeling Human-Object Interaction (HOI).** For HOI, we adopt an object-centric coordinate frame in which the object instance mesh $\mathcal{M}$ is centered at the origin. Our goal is to model the distribution of plausible human spatial and postural configurations relative to the object, covering a variety of HOI scenarios. Formally, we define HOI as the rotation $\mathbf{R}_\mathcal{H} \in \mathbb{R}^6$, translation $\mathbf{t}_\mathcal{H} \in \mathbb{R}^3$, scale

$\mathbf{s}_{\mathcal{H}} \in \mathbb{R}_+$, and body pose embedding $\theta_{\mathcal{H}} \in \mathbb{R}^H$ of a human $\mathcal{H}$. These are conditioned on an object mesh $\mathcal{M}$ and a textual description $\mathbf{c}$ describing the HOI. We denote the HOI distribution as $p_{\mathbf{c}}^{\mathcal{M}}$, and a corresponding HOI sample as $\phi^{\text{HOI}}$:

$$\phi^{\text{HOI}} \sim p_{\mathbf{c}}^{\mathcal{M}}, \quad \phi^{\text{HOI}} = (\mathbf{R}_{\mathcal{H}}, \mathbf{t}_{\mathcal{H}}, \mathbf{s}_{\mathcal{H}}, \theta_{\mathcal{H}}). \tag{1}$$

We use SMPL-X [39] for the human model and 6D representation [70] for $\mathbf{R}_{\mathcal{H}}$. Given an HOI sample $\phi^{\text{HOI}}$, human mesh $\mathcal{H}$ can be obtained as follows:

$$\begin{aligned} \mathcal{H} &= \mathbf{s}_{\mathcal{H}} \cdot \mathbf{R}_{\mathcal{H}} \mathcal{H}^{\text{cano}} + \mathbf{t}_{\mathcal{H}}, \\ \mathcal{H}^{\text{cano}} &= \mathbf{smplx}(\mathbf{dec}(\theta_{\mathcal{H}})), \end{aligned} \tag{2}$$

where $\mathbf{dec}(\cdot)$ is the body pose decoder (detailed below) that maps the body pose embedding $\theta_{\mathcal{H}}$ to SMPL-X pose $\theta \in \mathbb{R}^{21 \times 6}$. The function $\mathbf{smplx}(\theta)$ returns the SMPL-X mesh for pose $\theta$ in its canonical frame. We use by $\mathbf{R}_{\mathcal{H}}$ both the 6D representation and the corresponding SO(3) rotation, since there is a straightforward one-to-one mapping between them.

**Modeling Human-Human Interaction (HHI).** Given a text prompt $\mathbf{c}$ that describes the interaction between two humans, denoted $\mathcal{H}_1$ and $\mathcal{H}_2$, we model the HHI using their body poses along with the relative rotation and translation of $\mathcal{H}_2$ with respect to $\mathcal{H}_1$. We assume both humans share the same scale. Specifically, we define the HHI distribution as $p_{\mathbf{c}}^{\mathcal{H} \to \mathcal{H}}$ and the HHI sample as $\phi^{\text{HHI}}$:

$$\phi^{\text{HHI}} \sim p_{\mathbf{c}}^{\mathcal{H} \to \mathcal{H}}, \quad \phi^{\text{HHI}} = (\theta_{\mathcal{H}_1}, \mathbf{R}_{\mathcal{H}_2 \to \mathcal{H}_1}, \mathbf{t}_{\mathcal{H}_2 \to \mathcal{H}_1}, \theta_{\mathcal{H}_2}). \tag{3}$$

The human $\mathcal{H}_2$ can be reconstructed from the HOI of $\mathcal{H}_1$ and the HHI:

$$\begin{aligned} \mathcal{H}_2 &= \mathbf{s}_{\mathcal{H}_1} \cdot \mathbf{R}_{\mathcal{H}_1} \mathbf{R}_{\mathcal{H}_2 \to \mathcal{H}_1} \mathcal{H}_2^{\text{cano}} + \mathbf{s}_{\mathcal{H}_1} \cdot \mathbf{R}_{\mathcal{H}_1} \mathbf{t}_{\mathcal{H}_2 \to \mathcal{H}_1} + \mathbf{t}_{\mathcal{H}_1}, \\ \mathcal{H}_2^{\text{cano}} &= \mathbf{smplx}(\mathbf{dec}(\theta_{\mathcal{H}_2})), \end{aligned} \tag{4}$$

**Body Pose Embedding.** We find that modeling the human pose distribution using a low-dimensional embedding $\theta_{\mathcal{H}}$ is more effective than using the full 126D (=21 $\times$ 6, 6D representation for each joint rotation) body pose $\theta$ directly. Therefore, we train a body pose encoder and decoder to obtain an embedding vector of the body pose, enabling us to model HHOI in the latent space of body poses. To this end, we process 922K human body pose data from [18, 20, 52], and train a body pose encoder and decoder, each implemented as a 4-layer MLP. In experiments, we embed 126D human body poses in a 10D space, that is, $H = 10$, which results in $\phi^{\text{HOI}} \in \mathbb{R}^{20}$ and $\phi^{\text{HHI}} \in \mathbb{R}^{29}$.

## 3.2 Score-based HHOI Diffusion Model

We model HHOI using score-based diffusion, similar to how poses and scales of objects are modeled in [1, 62]. Let us denote our HHOI diffusion model as $\Psi_{\Theta}$, parameterized by $\Theta$. Then, $\Psi_{\Theta}$ represents the noised score function of the HOI, HHI distribution at time step $t$:

$$\Psi_{\Theta^z}(\phi_t^z, t | \mathbf{c}, z) = \begin{cases} \nabla_{\phi_t^{\text{HOI}}} \log p_{\mathbf{c}}^{\mathcal{M}}(\phi_t^{\text{HOI}}), & z = \text{HOI} \\ \nabla_{\phi_t^{\text{HHI}}} \log p_{\mathbf{c}}^{\mathcal{H} \to \mathcal{H}}(\phi_t^{\text{HHI}}), & z = \text{HHI} \end{cases}, \tag{5}$$

where $\phi_t^{(\cdot)}$ is a noised HOI or HHI sample at time step $t$, $z \in \{\text{HOI}, \text{HHI}\}$ represents mode selector, and $\Theta^{\text{HOI}} \cup \Theta^{\text{HHI}} = \Theta$, $\Theta^{\text{HOI}} \cap \Theta^{\text{HHI}} = \emptyset$. For simplicity, we do not use a mesh instance $\mathcal{M}$ as input when modeling HOI; rather, we assume a fixed $\mathcal{M}$ is provided for each scenario.

**Training.** As HHOI is formulated by decomposing it into HOI and HHI in Sec. 3.1, we also train the score-based diffusion in a decoupled manner. Each mode is trained with the following objective function presented in Denoising Score Matching(DSM) [51]:

$$\mathcal{L}_{\text{score}}(\Theta^z) = \mathbb{E}_{t \sim \mathcal{U}(\epsilon, 1)} \left[ \lambda_t \mathbb{E}_{\phi^z, \phi_t^z} \left[ \left\| \Psi_{\Theta^z}(\phi_t^z, t | \mathbf{c}, z) - \frac{\phi^z - \phi_t^z}{\sigma(t)^2} \right\|_2^2 \right] \right], \tag{6}$$

where $\epsilon$ is minimal noise level, $\phi_t^z \sim \mathcal{N}(\phi^z, \sigma^2(t) \mathbf{I}_z)$, $\sigma^2(t) = \sigma_{\min}(\frac{\sigma_{\max}}{\sigma_{\min}})^t$ is a variance factor, and $\lambda_t$ is a regularization term.

The training process is shown in Fig. 2(a). First, we generate perturbed samples $\phi_t^z$ as defined. We then compute the target score $\frac{\phi^z - \phi_t^z}{\sigma(t)^2}$ and the estimated score from the HHOI diffusion model to

evaluate the score loss in Eq. (6). We use the CLIP text encoder [40] to obtain a text embedding from a text prompt. For simplicity, we use $\mathbf{c}$ to denote both the text prompt and its corresponding embedding. Additionally, we adopt the text prompt augmentation strategy from [1].

**Inference.** To sample HOI and HHI instances independently, we solve the following Probability Flow(PF) ODE [48] in reverse time, i.e., from $t = 1$ to $t = \epsilon$:

$$\phi_1^z \sim \mathcal{N}(\mathbf{0}, \sigma_{\max}^2 \mathbf{I}_z),$$
$$\frac{d\phi_t^z}{dt} = -\sigma(t)\dot{\sigma}(t)\Psi_{\Theta^z}(\phi_t^z, t | \mathbf{c}, z). \tag{7}$$

We solve the ODE using external library [7], which is fully supported to run on the GPU. Fig. 2(a) shows this process. However, independently sampling HOI and HHI leads to incoherent posture and spatial configuration in each sampling output, and does not give us the desired HHOI. To address this issue, we propose an advanced guided sampling technique to obtain HHOI in Sec. 3.3.

### 3.3 Advanced Guided Sampling for HHOI

We adopt the PF ODE augmentation paradigm [1, 27], introducing additional terms that enforce the appropriate combination of HOI and HHI samples during HHOI sampling. Specifically, we incorporate an inconsistency loss to enforce consistency across samples and a collision loss to prevent human collisions. Consider the HHOI involving $N$ humans(i.e., $N$ HOIs), along with $M$ HHIs:

$$\phi^{\text{HOI}, \mathcal{H}_i} \sim p_{\mathbf{c}^{\text{HOI}}}^{\mathcal{M}}, \quad \phi^{\text{HOI}, \mathcal{H}_i} = (\mathbf{R}_{\mathcal{H}_i}, \mathbf{t}_{\mathcal{H}_i}, \mathbf{s}_{\mathcal{H}_i}, \theta_{\mathcal{H}_i}),$$
$$\phi^{\text{HHI}, \mathcal{H}_k \to \mathcal{H}_j} \sim p_{\mathbf{c}^{\text{HHI}}}^{\mathcal{H} \to \mathcal{H}}, \quad \phi^{\text{HHI}, \mathcal{H}_k \to \mathcal{H}_j} = (\theta_{\mathcal{H}_j}, \mathbf{R}_{\mathcal{H}_k \to \mathcal{H}_j}, \mathbf{t}_{\mathcal{H}_k \to \mathcal{H}_j}, \theta_{\mathcal{H}_k}), \tag{8}$$

where $i = 1, \ldots, N$, and $|\{(j, k) \mid 1 \le j, k \le N, j \ne k\}| = M \le \frac{N(N-1)}{2}$. Note that the number of possible HHIs is at most $_N C_2 = \frac{N(N-1)}{2}$. The graph formed by connecting the $M$ human pairs should be a directed acyclic graph (DAG). For example, given three HHIs $\{\mathcal{H}_2 \to \mathcal{H}_1, \mathcal{H}_3 \to \mathcal{H}_2, \mathcal{H}_1 \to \mathcal{H}_3\}$, this is not a valid HHI set because it has a circular dependency.

**Inconsistency Loss.** To generate a unified HHOI sample from inconsistency HOI/HHI samples set, we introduce inconsistency loss $\mathcal{L}_{\text{inc}}$ enforcing coherence between the human representations derived by each sample. In a nutshell, $\mathcal{L}_{\text{inc}}$ is the role of enforcing consistency between humans obtained from Eq. 2 and humans obtained from Eq. 4. Specifically, it penalizes discrepancies in scale, body pose, rotation, and translation of HOI and HHI samples at time step $t$ during sampling, as follows:

$$\mathcal{L}_{\text{inc}}(\Phi_t) = \mathcal{L}_{var,s}(\mathbf{s}) + \mathcal{L}_{var,\theta}(\theta) + \mathcal{L}_{var,R}(\mathbf{R}) + \mathcal{L}_{var,t}(\mathbf{t}), \tag{9}$$

where $\Phi_t$ denotes union of $N$ HOI samples $\{\phi_t^{\text{HOI}, \mathcal{H}_i}\}$ and $M$ HHI samples $\left\{\phi_t^{\text{HHI}, \mathcal{H}_k \to \mathcal{H}_j}\right\}$ at time step $t$. Each term in Eq. (9) minimizes the variance of each component, thereby enhancing overall consistency as detailed below.

The scale variance loss, $\mathcal{L}_{var,s}(\mathbf{s})$ penalizes deviations in human scale across HOI samples since the HHI model assumes equal scales for paired humans:

$$\mathcal{L}_{var,s}(\mathbf{s}) = N \cdot \text{Var}(\{\mathbf{s}_{\mathcal{H}_i}\}_{i=1}^N). \tag{10}$$

The body pose variance loss $\mathcal{L}_{var,\theta}(\theta)$ enforces consistency in body poses for the same human:

$$\mathcal{L}_{var,\theta}(\theta) = \sum_{i=1}^N N_i \cdot \text{Var}(\{\theta_{\mathcal{H}_i,n}\}_{n=1}^{N_i}), \tag{11}$$

where $N_i$ denotes total occurrences of human $\mathcal{H}_i$ across HOI and HHI samples. More precisely, $N_i - 1$ is the number of HHIs where $\mathcal{H}_i$ appears. The rotation and translation variance losses, $\mathcal{L}_{var,R}(\mathbf{R})$ and $\mathcal{L}_{var,t}(\mathbf{t})$, ensure consistency among human rotation and translation across interactions:

$$\mathcal{L}_{var,R}(\mathbf{R}) = \sum_{i=0}^N N_i' \cdot \text{Var}(\{\mathbf{R}_{\mathcal{H}_i}\} \cup \{\mathbf{R}_{\mathcal{H}_{jn}} \mathbf{R}_{\mathcal{H}_i \to \mathcal{H}_{jn}}\}_{n=1}^{N_i'}), \tag{12}$$

$$\mathcal{L}_{var,t}(\mathbf{t}) = \sum_{i=0}^N N_i' \cdot \text{Var}(\{\mathbf{t}_{\mathcal{H}_i}\} \cup \{\mathbf{s}_{\mathcal{H}_{jn}} \cdot \mathbf{R}_{\mathcal{H}_{jn}} \mathbf{t}_{\mathcal{H}_i \to \mathcal{H}_{jn}} + \mathbf{t}_{\mathcal{H}_{jn}}\}_{n=1}^{N_i'}), \tag{13}$$

where $N_i'$ represents the number of HHIs where human $\mathcal{H}_i$ appears as a target(i.e., in pairs of the form $\mathcal{H}_i \to \mathcal{H}_j$). Note that $\mathrm{Var}(\{x_1, \ldots, x_n\}) = \frac{1}{n} \sum_{i=1}^{n} (x_i - \bar{x})^2$.

**Collision Loss.** To ensure plausible interactions among multiple humans, it is essential to avoid physically implausible scenarios such as unintended body intersections. In particular, for human pairs that are not explicitly connected by a HHI, we assume an "no collision" constraint—that is, the absence of collision serves as their implicit HHI. For instance, if the HHI set is $\{\mathcal{H}_2 \to \mathcal{H}_1, \mathcal{H}_3 \to \mathcal{H}_2\}$, then we enforce that $\mathcal{H}_1$ and $\mathcal{H}_3$ do not collide.

However, computing accurate inter-human collisions using full SMPL-X meshes during ODE-based sampling is computationally expensive. To address this, we approximate collisions between two humans in following steps: (1) Compute joint positions via forward kinematics over body joints using each human pose; (2) Construct a 24-capsule proxy from these joints with predefined radius factors to approximate each human as a capsule-based model; (3) Compute the sum of overlaps over all capsule pairs for two humans as the collision loss. Each overlap is computed simply as the sum of the two radii minus the distance between the capsules' axis segments. Accordingly, the collision loss $\mathcal{L}_{\mathrm{col}}(\Phi_t)$ is computed as

$$\mathcal{L}_{\mathrm{col}}(\Phi_t) = \sum_{(\mathcal{H}_i, \mathcal{H}_j) \in \Phi_{\mathrm{nap}}} \frac{1}{24^2} \sum_{c_i=1}^{24} \sum_{c_j=1}^{24} \max(0, r_{c_i}^{\mathcal{H}_i} + r_{c_j}^{\mathcal{H}_j} - d_{c_i,c_j}^{\mathcal{H}_i,\mathcal{H}_j}), \tag{14}$$

where $\Phi_{\mathrm{nap}}$ is the set of non-adjacent human pairs, $r_{c_i}^{\mathcal{H}_i}$ is the radius of the $c_i$-th capsule of human $\mathcal{H}_i$, and $d_{c_i,c_j}^{\mathcal{H}_i,\mathcal{H}_j}$ is the distance between the axis segment of the $c_i$-th capsule of human $\mathcal{H}_i$ and the axis segment of the $c_j$-th capsule of human $\mathcal{H}_j$. Note that in Eq. 14, if the sum of the two capsule radii is less than the distance between their axis segments, there is no collision and the value is clipped to zero. See Supp. Mat. for a process to approximate a human with a 24-capsule proxy.

**Guided HHOI Sampling.** Using Eq. 9 and Eq. 14, we augment the PF ODE in Eq. 7 as follows to sample HHOI:

$$\phi_1^{z,i} \sim \mathcal{N}(\mathbf{0}, \sigma_{\max}^2 \mathbf{I}_z),$$

$$\frac{d\phi_t^{z,i}}{dt} = -\sigma(t)\dot{\sigma}(t)\Psi_{\Theta^z}(\phi_t^{z,i}, t | \mathbf{c}^z, z) + \lambda_1 \nabla_{\phi_t^{z,i}} \mathcal{L}_{\mathrm{inc}}(\Phi_t) + \lambda_2 \nabla_{\phi_t^{z,i}} \mathcal{L}_{\mathrm{col}}(\Phi_t), \tag{15}$$

where $\phi_t^{z,i}$ is $i_{\mathrm{th}}$ HOI($z$ = HOI) sample or HHI($z$ = HHI) sample at time step $t$, $\lambda_1$ and $\lambda_2$ are weight terms. In Fig. 2(b), we provide an intuitive overview of our advanced sampling process through a concrete HHOI example in which three humans sit on a bench.

## 4 Datasets

In this section, we present the motivation and data collection procedure for our dataset. To align with our model architecture, the distribution of body poses in the HOI dataset must be consistent with that of the HHI dataset. For instance, HOI dataset with action "sitting on a bench", may consist only of body pose with head pose fixed to the front, assuming only one human is in the scene, while the HHI dataset may contain a broader range of head orientations more representative of real-world variability. In this case, our model will be unable to generate these natural variations during the HHOI synthesis.

The most straightforward solution is to collect data from scenes that include both multiple humans and interacting objects. While several individual datasets that contain either multi-human poses or human-object interactions have been released, there are a limited number of datasets that capture both multi-human poses and human-object interactions. CORE4D [37] is one of them, providing high-quality motion capture data of two individuals interacting with a single object. It includes object pose annotations and human SMPL-X parameters for six object categories. However, the range of interaction types in CORE4D is relatively narrow, focusing primarily on collaborative actions such as passing an object or moving it together. In contrast, real-world interactions are more diverse. Notably, there are scenarios where no direct contact occurs between humans and objects, yet the mere presence of the object influences spatial relationships (proxemics) and interaction dynamics between individuals. Such implicit object effects are underrepresented in existing datasets.

To address the lack of diverse human-human-object interactions, we collect dyadic human and object poses using a multi-view camera system. Specifically, we record two people interacting with various

objects from 36 synchronized RGB cameras. We apply an off-the-shelf 2D pose detector [59] and a 3D optimization method [69] to recover the SMPL-X parameters for both individuals in each camera frame, followed by manual post-filtering. The collected HHOI dataset is divided into HOI and HHI subsets, which are used to train the respective score-based models. The same data processing and training procedure is applied to the CORE4D dataset. Our dataset consists of 5,078 frames spanning 11 object categories, and is expressed in metric units. Refer to Supp. Mat. for more details.

While the aforementioned datasets provide high-quality HHOI data, their content is limited to scenarios that can be captured in controlled studio environments. As a result, interactions involving large objects or outdoor scenarios, such as "two people riding a motorcycle together", are difficult to capture with our multi-view camera systems. To address this limitation, we leverage the knowledge of 2D image diffusion model on HOI and HHI. Recent models such as Flux [31] can generate realistic human-object interaction images from textual prompts while preserving crucial social cues, including joint body poses and interpersonal distances that reflect real-world dynamics.

To generate HOI data, we utilize ComA [26], which takes as input an object instance and a textual description of an HOI scenario, and outputs corresponding 3D human-object interactions represented via SMPL-X parameters. For HHI data, we generate images of dyadic human interactions involving an object using textual prompts and the Flux diffusion model. Subsequently, we apply a Human Mesh Recovery method [2] to reconstruct 3D human meshes from the generated images.

## 5 Experiments

### 5.1 Baselines and Metrics

**Baselines.** Previous research on human-object interaction has primarily focused on generating a single human pose with respect to a specific object. As there exists no comparable method for generating multiple humans interacting with an object as in our setting, we develop two approaches for this novel task: (1) by extending GenZI [33], and (2) by lifting separately generated humans using a depth optimization strategy, referred to as Depth Opt.

First, we modify GenZI, which originally renders a target object in multi-view and inpaints a single human based on a text prompt. We revise the prompt and inpainting process to synthesize images with multiple people, and follow GenZI's pipeline by detecting 2D body poses and optimizing SMPL-X parameters using multi-view joint positions. For Depth Opt., we inpaint a single-view object rendering using a diffusion-based model [41], extract 3D human meshes via Human Mesh Recovery [2], and align them to the object using depth optimization with Depth-Pro [5]. See Supp. Mat. for further details.

**Metrics.** To evaluate how realistically our model generates body poses and interpersonal distances, we compare the distributions of generated results against the test set in CORE4D and our collected dataset. For body pose, we compute the Fréchet Distance (FD) between the embedded pose distributions which are obtained via our body pose encoder. For interpersonal distance, we calculate FD on the per-human global translation differences.

We additionally employ CLIP-score [40] to assess semantic alignment between the generated outputs and the input text prompts. The HHOI outputs are rendered from multi-view, and CLIP-score is computed by averaging image-text cosine similarities across these views. For cases involving more than three humans, we report the success rate to evaluate the robustness of our model in multi-human settings. We account generation success as fully generating desired number of people in 3D.

Physical plausibility of the generated HHOI is also an important factor, so we evaluate it using two metrics—penetration ratio and contact distance. Penetration ration measures the ratio of mesh vertices that lie inside another mesh, against the total mesh vertices. We count human vertices inside another human or the object and report human-human and human-object separately. Contact distance is measured only for categories requiring contact, split into hand-contact (board, box, bucket, chair, desk) and hip-contact (bench). For each, we randomly sample 10 vertices from the relevant body parts and compute their mean distance to the object mesh.

Finally, we conduct a user study to further validate the realism and text coherence of our method. Participants compare outputs from our model and baselines and select the most realistic and faithful to the prompt.

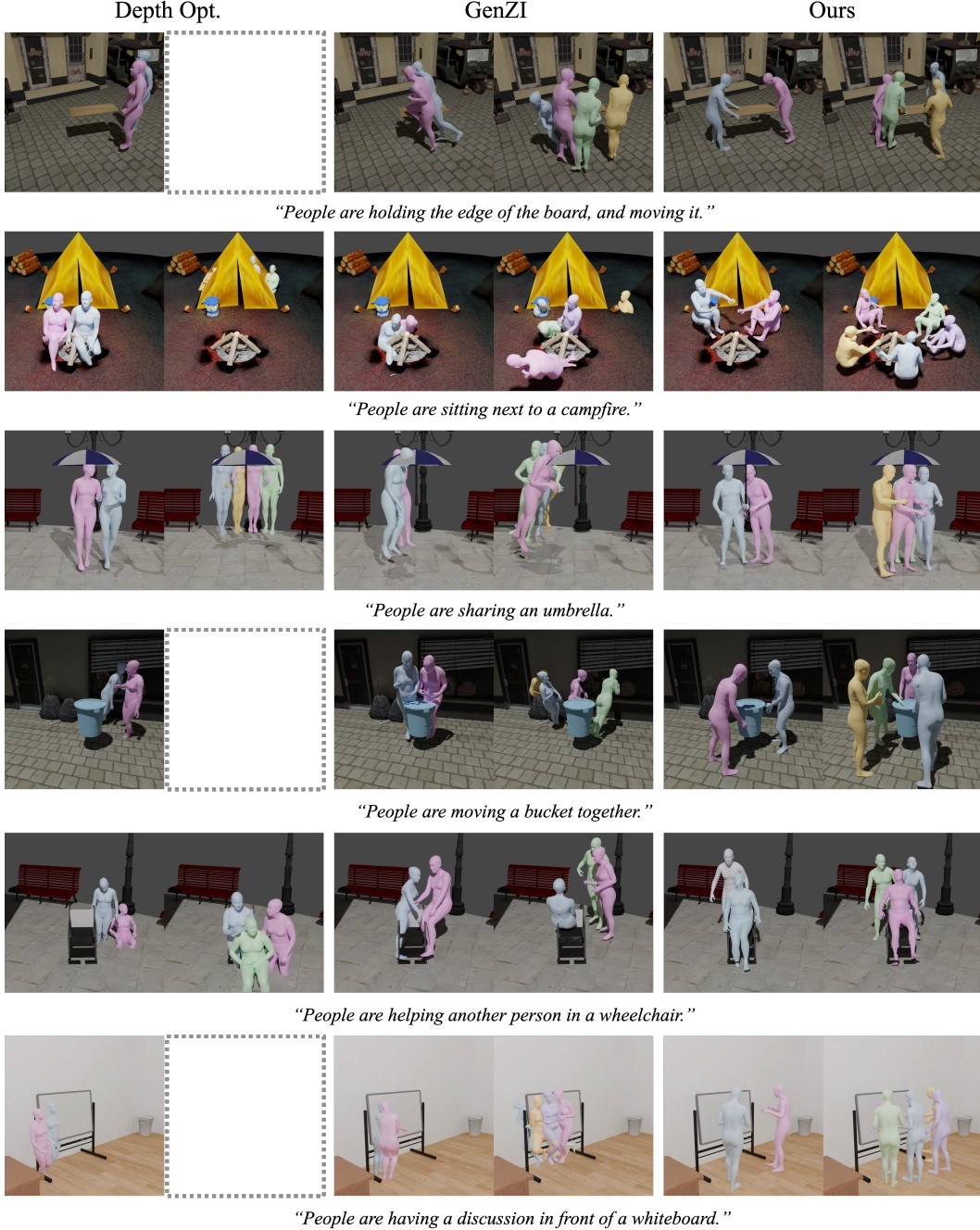

| Depth Opt. | GenZI | Ours |

*"People are holding the edge of the board, and moving it."*

*"People are sitting next to a campfire."*

*"People are sharing an umbrella."*

*"People are moving a bucket together."*

*"People are helping another person in a wheelchair."*

*"People are having a discussion in front of a whiteboard."*

Figure 3: HHOI generation result of dyadic, and multiple humans in action with our model and baselines. In multiple HHOI, number of humans ranges from 3 to 5. Empty result represents cases where generation failed in 10 trials. Our model can generate complex HHOIs with varying number of humans in the scene, while preserving the natural social cues.

## 5.2 Results

Qualitative results on our model output, along with baseline outputs is shown in Fig. 3. Since both baseline models leverage the knowledge of 2D image diffusion model to generate plausible HOIs, their performance depends heavily on the inpainting output. In scenes where the object is actively used and in direct contact with a human, inpainting quality degrades; the target object is frequently missing from the output, which in turn leads to poor optimization quality. Additionally, as the number of people increases, the percentage of inpainting where it fails to generate the whole $N$ number of

Table 1: Quantitative comparison on text-guided dyadic HHOI generation. Our model shows robust performance compared to baseline models in all the metrics.

| Method | Body Pose FD ↓ | Distance FD ↓ | CLIP Score ↑ | User Study (%) |
|---|---|---|---|---|
| Depth Opt. | 0.6834 | 0.4180 | 0.2647 | 20.8 |
| GenZI | 1.3500 | 0.3542 | 0.2633 | 18.4 |
| **Ours** | **0.1755** | **0.0689** | **0.2695** | **60.9** |

Table 2: Quantitative comparison of text-guided multi-human generation. Our model consistently outperforms baseline methods across all evaluation metrics, demonstrating robust performance.

| Method | CLIP Score ↑ | Success Rate (%) | | |
|---|---|---|---|---|
| | | 3 human | 4 human | 5 human |
| Depth Opt. | 0.2510 | 33.3 | 18 | 13 |
| GenZI | 0.2584 | 67.3 | 41.4 | 22 |
| **Ours** | **0.2685** | **100.0** | **100.0** | **100.0** |

people also increases. The multi-view consistency of each human also further deteriorates as the number of people increases.

Aside from inpainting quality, both models suffer from lack of 3D HOI and HHI knowledge. Depth Opt. generates implausible human position relative to the object, due to depth estimation error. The multi-view SMPL-X parameter optimization in GenZI is sensitive to consistent body pose in each view, and can lead to poor body pose output quality. On the contrary, our generation results show high quality HHOIs in various object categories in both static and dynamic scenes.

Tab. 1 shows quantitative results evaluating the realism on generating two people in action with object. Compared to baseline models, our model achieves significantly higher score in body pose and distance FD, implying our model can produce more realistic and natural HHOIs, that resemble those in real world environment. Note that due to background scene rendering, images with different HHOI generation output may have similar CLIP embeddings, which may be the cause of all 3 models producing CLIP scores in close range. This suggests FD of body pose and interpersonal distance is a more reliable method in evaluating how realistic each output is. Nevertheless, our model beats the baseline model in CLIP score by a slight margin. Our model also outperforms baseline models in multi-human generation. Tab. 2 shows that our model achieves higher scores in CLIP score and success rate when generating more than 3 people, implying a more realistic and robust generation.

We show the physical plausibility metrics of our model output against the baselines in Tab. 3. Our model outperforms baseline models in contact distance metric, implying our model output achieves more precise contact with the object compared to baseline. For the human-human and human-object penetration ratios, our model demonstrates superior performance in dyadic human generation, whereas Depth Opt. achieves comparable results when generating three or more people. However, this apparent performance stems from Depth Opt.'s tendency to place humans at excessively large distances from objects, a consequence of the instability in depth estimation. When combined with Human Mesh Recovery, this results in low penetration metrics but unrealistic spatial arrangements in HHOI scenarios. The poor contact distance scores of Depth Opt. further substantiate this observation.

## 5.3 Applications

The capability of our model to generate plausible human-human-object interactions (HHOIs) involving multiple individuals from textual prompts provides a foundation for a range of future research directions. One promising avenue is generating multiple human motion in the presence of objects in the scene. In this way, we leverage our sampled human configurations as conditioning inputs for motion-in-betweening tasks.

Diffusion-Noise-Optimization (DNO) [25] performs various motion-related editing tasks, using existing motion-diffusion models as a motion prior. The output of naive motion generation is compared with the condition to calculate joint loss. By changing the sampling process to ODE, we

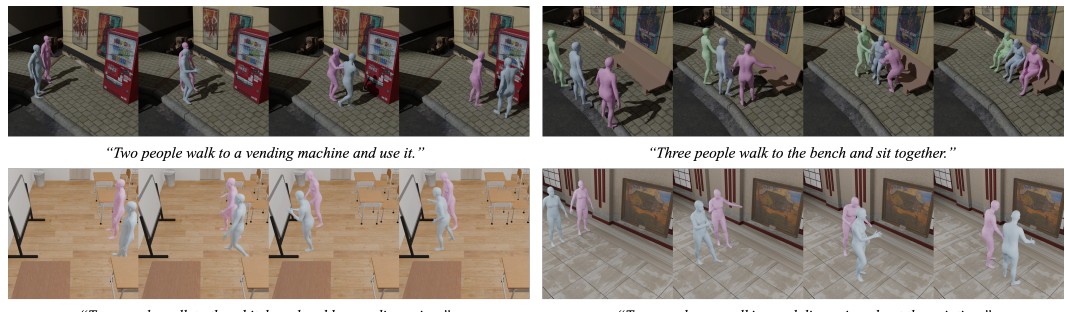

*"Two people walk to a vending machine and use it."*      *"Three people walk to the bench and sit together."*

*"Two people walk to the whiteboard and have a discussion."*      *"Two people are walking and discussing about the painting."*

Figure 4: Motion in-betweening outputs from DNO and InterGen, given a naive standing pose as the start frame constraint and our HHOI generation output as the end frame constraint.

Table 3: Quantitative comparison on physical plausibility metrics of our model and baseline outputs.

| Metric | Method | 2 Human | 3 Human | 4 Human | 5 Human |
|---|---|---|---|---|---|
| Human-Human Penetration Ratio ↓ ($\times 1000$) | Depth Opt. | 13.91 | **14.06** | **19.35** | 26.16 |
| | GenZI | 13.91 | 24.60 | 21.55 | 49.06 |
| | **Ours** | **7.90** | 15.68 | 19.82 | **25.69** |
| Human-Object Penetration Ratio ↓ ($\times 1000$) | Depth Opt. | 11.48 | **4.15** | 9.41 | **1.20** |
| | GenZI | 60.71 | 15.64 | 11.93 | 8.91 |
| | **Ours** | **5.49** | 7.74 | **7.19** | 10.18 |
| Contact Distance ↓ (m) | Depth Opt. | 0.666 | 1.992 | 4.415 | 2.736 |
| | GenZI | 0.103 | 0.126 | 0.537 | 0.389 |
| | **Ours** | **0.029** | **0.031** | **0.028** | **0.026** |

can obtain a motion sample from the latent noise deterministically and retrieve the latent noise from the motion output. Since this process is deterministic, we can backpropagate the joint loss to the inversion process and optimize the latent noise to minimize the joint loss.

Human samples generated by our model are utilized as conditional inputs to guide the optimization of the motion diffusion model InterGen [35]. By integrating object interaction cues from our sampled HHOIs with motion priors from diffusion-based models, we enable the synthesis of natural multi-human motions that exhibit plausible interactions with objects in the environment. As illustrated in Fig. 4, our method allows precise positioning of generated humans and object-specific poses that are not observed in existing models.

## 6 Discussion

In this paper, we propose a method to model Human-Human-Object Interactions (HHOIs) using score-based generative models. By combining separately trained HOI and HHI models, we introduce a novel sampling strategy that enables the generation of an arbitrary number of people interacting with an object. To train and evaluate our model, we construct a new HHOI dataset by capturing additional samples, together with synthetic data generated by our pipeline. We demonstrate that our method can synthesize realistic multi-human interaction with diverse objects. As shown in our applications, this capability enables downstream tasks such as interaction-aware motion generation and provides a foundation for future research in multi-agent embodied intelligence.

As a limitation, our score-based model cannot directly learn HHOIs from datasets dedicated solely to HOI or HHI, due to distributional discrepancies in human configurations. Extending our framework to effectively leverage such datasets remains an open research question.

## Acknowledgements

This work was supported by NRF grant funded by the Korean government (MSIT) [No. RS-2022-NR070498 and RS-2025-25396144], and IITP grant funded by the Korea government (MSIT) [No. RS-2024-00439854, No. RS-2025-25441838, No. RS-2021-II211343, No. RS-2025-25442338, and No.2022-0-00156]. H. Joo is the corresponding author.

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
