# OpenReview forum: "Learning to Generate Human-Human-Object Interactions from Textual Descriptions"
_NeurIPS.cc/2025/Conference — NeurIPS 2025 poster_

### Official Review · Reviewer_hSov · 2025-06-16

**Clarity:** 3
**Significance:** 3
**Originality:** 3
**Rating:** 4
**Confidence:** 3

**Summary:**

This paper introduces Human-Human-Object Interactions (HHOI) to model people interacting around an object. It introduces a new dataset and a score-based diffusion model to generate realistic multi-person scenes from object and text inputs, outperforming single-person HOI methods and extending to multi-person interactions.

**Questions:**

n/a

**Ethical Concerns:**

["NO or VERY MINOR ethics concerns only"]

**Final Justification:**

The authors have addressed most of my concerns, I recommend borderline accept.

**Limitations:**

I believe HHOI has unique value for multiple people interaction in complex scenarios, especially in embodiment AI. Although it raises concerns about potential biases in synthetic data, the novel HHOI framework and generative approach warrant recognition.

**Quality:**

3

**Strengths And Weaknesses:**

Strengths
1. The proposal of HHOI generation is novel.

2. Develops a novel real-world HHOIs dataset and employs image generative models for synthetic data, enabling coverage of dyadic to multi-person interaction scenarios.

3. Utilizes a score-based diffusion framework to merge HOI and HHI modalities, allowing end-to-end generation of interactions from object inputs and text descriptions, scalable from 2 to multiple humans.

Major Weaknesses:

1. Since HHOI is composed of HOI and HHI, it is essential to respectively compare the HOI and HHI results generated by HHOI with those from dedicated HOI and HHI methods. For HHI, previous approaches suitable for comparison include diffusion-based methods such as ComMDM (ICLR 2024), InterGen (IJCV 2024), in2IN (CVPRW 2024), and the transformer-based InterMask (ICLR 2025). For HOI, InterDiff(ICCV 2023), HOI-Diff(CVPRW 2025), CHOIS(ECCV 2024) can be selected for comparation. Metrics about generated motion quality such as R-precision, MM-Dist, MModality and Diversity can be evaluated. Interaction quality metrics mentioned in CHOIS also can be used for HHOI for evaluation. Additionally, can the HOI diffusion and HHI diffusion be replaced with other generative models? For example, will replacing the generated sequence from HHI diffusion with InterMask or ComMDM improve quality?

2. Although the advanced sampling technique includes collision loss to minimize human-body intersections, in figure 3, meshes still exhibit unnatural interaction, particularly in hand movements.

Minor Weaknesses:
1. Synthetic data may introduce bias due to differences from real-world environments, as pretrained models may ignore spatial coherence or collision constraints in 3D space. The real-world dataset is predominantly dyadic, with multi-person synthetic data (3–5 people) relying on generated priors. This may affect robustness in complex scenarios.

2. The paper does not explicitly state the proportion of synthetic data to real data in the dataset.

3. When multiple people interact simultaneously, such as three people stacking their hands together when they touch the object, can HHOI generate such complex scenarios?

---

> ### Author Rebuttal · Authors · 2025-07-27
>
> **Q1) Compare our HOI and HHI results with dedicated HOI and HHI methods**
>
> We clarify that our primary contribution of our paper lies in generating static HHOI poses, not dynamic HHOI motion generation. However, since we proposed HHOI motion generation as an application in Sec. 5, it is also meaningful to compare it with the motion generation methods. We perform additional experiments and evaluations on motion generation following the reviewer's guidance.
>
> We generate HHOI motions using the method proposed in Sec. 5 and then split each sequence into an HHI motion and HOI motions. Because a single HHOI motion can yield two HOI motions, we randomly select one of them. Then, we compare them with ComMDM and InterGen for HHI, and HOI-Diff and InterDiff for HOI.
>
> We report the *Diversity*, *MM-Dist*, and *MModality* as evaluation metrics. *Diversity* measures the total diversity of the generated motions by computing the average distance between 300 random sampled motion pairs. *MM-Dist* is the average Euclidean distance between the input text embedding and the output motion embedding.  *MModality* similarly measures diversity between motions generated from the same text prompt.
>
> For HHI motion, we evaluate on 16 object categories. For HOI motion, we evaluate on 4 object categories—box, chair, desk, and monitor—because these categories are shared by our dataset and the BEHAVE dataset on which HOI-Diff and InterDiff were trained. In both cases, we generated 10 scenes per object category to compute metrics.
>
> *Table 1. Quantitative evaluation of HHI motion against existing methods.*
>
> | Method   |  Diversity ($\uparrow$) | MM-Dist ($\downarrow$) | MModality ($\uparrow$) |
> |---------------|--------:|--------:|--------:|
> | ComMDM | 1.70   | 4.10  | **1.26** |
> | InterGen |   **1.82** |  **3.82**   | 1.02 |
> | Ours | 1.73 |  4.00 | 1.26 |
>
> ---
>
> *Table 2. Quantitative evaluation of HOI motion against existing methods.*
>
> | Method       |  Diversity ($\uparrow$) | MM-Dist ($\downarrow$) | MModality ($\uparrow$) |
> |---------------|--------:|--------:|--------:|
> | HOI-diff |  **5.09**   | **16.06**  | **3.39** |
> | InterDiff |  4.84  |  16.68  | 3.08 |
> | Ours | 3.34 | 16.12  | 2.89 |
>
> ---
>
> Our results are comparable to those of existing motion generation methods. For HHI motion generation, our method outperforms ComMDM in Diversity and MM-Dist and surpasses InterGen in MModality. For HOI motion generation, we achieve a better MM-Dist than InterDiff.
>
> **Q2) Can the HOI diffusion and HHI diffusion be replaced with other generative models?**
>
> They cannot be replaced with other types of models. We emphasize that our contribution also lies in modeling HOI and HHI using a score-based diffusion approach, since these models serve as the components enabling coherent HHOI sampling. While previous methods have addressed these two tasks independently, merging the two individual models into a single model has been a challenge due to each model's unique structure.
>
> **Q3) Unnatural hand movements**
>
> Our work focuses on coordinated interaction among multiple humans for a single object, rather than precise hand-object contact. So, we only model SMPL-X body poses and do not include hand poses. The unnatural hand movements stem from this design choice. As future work, we plan to incorporate hand modeling to address this issue.
>
> **Q4) Regarding the phrase "multi-person synthetic data (3–5 people)"**
>
> We emphasize that all our datasets (including the captured data, CORE-4D, and synthetic data) consist solely of dyadic scenarios. The ability of our HHOI diffusion model to generate multi-human (>2) interactions does not stem from the presence of multi-human data, but rather from our advanced sampling strategy.
>
> **Q5) Synthetic data may introduce bias**
>
> Our use of synthetic data is limited to scenarios that are difficult to capture in studio settings (e.g., two people riding a motorcycle together), as mentioned in Sec. 4.
>
> **Q6) The proportion of synthetic data to real data**
>
> Dataset we used in training consists of data from CORE-4D, multi-view capture system, and from our synthetic pipeline. A total of 4080 HOI and 4080 HHI samples are used from CORE-4D, 9430 HOI and 9430 HHI samples from our multi-view capture system, and 159 HOI and 140 HHI samples from our synthetic pipeline. That is, our dataset consists of 99% real data and 1% synthetic data.
>
> **Q7) Can HHOI diffusion generate scenarios where multiple people interact simultaneously?**
>
> It is possible, but as mentioned in **Q3**, precise hand poses are not modeled.

---

> > ### Comment · Reviewer_hSov · 2025-08-05
> >
> > The authors addressed most of my concerns, I will keep my rating.

---

### Official Review · Reviewer_w3tN · 2025-06-26

**Clarity:** 3
**Significance:** 2
**Originality:** 3
**Rating:** 4
**Confidence:** 4

**Summary:**

This paper proposes a method to generate human-human-object interactions (HHOIs), which is a novel and challenging task in the community. The authors decompose the task into human-human interaction (HHI) and human-object interaction (HOI), and propose an advanced sampling strategy with an inconsistency loss and a collision loss to combine the results. To train the model to generate more natural HHOIs, the authors also build a new dataset captured using multi-view systems. The qualitative results demonstrate that the method can produce plausible HHOIs.

**Questions:**

Please address the concerns raised in the weaknesses section, particularly those related to the sampling strategy, loss functions, training process, dataset, and evaluation metrics.

**Ethical Concerns:**

["NO or VERY MINOR ethics concerns only"]

**Final Justification:**

I believe most of concerns from other reviewers are also addressed. The dataset might be useful for the community.

**Limitations:**

yes

**Paper Formatting Concerns:**

N.A

**Quality:**

3

**Strengths And Weaknesses:**

Strengths
- Decomposing HHOI into HOI and HHI reduces the training difficulty and enables the generation of arbitrary individuals.
- The proposed Advanced Sampling strategy considers how to integrate multiple results and constrains their interrelations to form a coherent HHOI.
- The proposed dataset may be valuable for future research on HHOI.


Weaknesses
- The summary of contributions may need to be reorganized. This paper indeed has meaningful contributions, such as the decomposition of HHOI, the joint sampling of HOI and HHI, and the proposed dataset. However, the current summary does not accurately reflect these contributions. For example, Text-to-HOI and Text-to-HHI should not be claimed as contributions of this paper, as the authors did not make any technical improvements in this area.
- What is the fundamental difference between HHOI and prior HHI tasks? The authors should clearly articulate this in the introduction. Since HHOI is a relatively new task, clarifying the distinctions would help readers better understand its importance and highlight the contributions of this work.
- Why not use VPoser [A] for pose embedding? Both approaches essentially aim to achieve the same goal.

     [A] "Expressive body capture: 3d hands, face, and body from a single image." Proceedings of the IEEE/CVF conference on computer vision and pattern recognition. 2019.
- Why is scale in the human model introduced? Why not normalize all input objects to a unified scale instead? This could simplify the problem and better align with real-world applications.
- What is the exact formulation of the variance loss? A clear mathematical description is necessary.
- The proposed method samples N + N(N–1) humans. How is the final result selected from these candidates?
- How is the directed acyclic graph (DAG) constructed? How does the model determine which individuals are involved in HHI with each other?
- Does combining HOI and HHI to form HHOI require further training? What datasets are used to train the HOI and HHI components respectively?
- Collision loss cannot capture fine-grained interactions. Since some interactions may not involve mesh-level penetration but could still result in overlapping bounding boxes, could the collision loss penalize these correct interactions?
- The constructed real-world dataset contains only 5,078 frames, which is relatively small in scale. Moreover, current reconstruction methods cannot accurately model collaborative interaction scenarios, which may result in low-quality generated data. How much does this dataset actually contribute to the improvement of the proposed method?
- The current evaluation metric, Fréchet Distance (FD), is insufficient to assess the quality of interactions.

---

> ### Author Rebuttal · Authors · 2025-07-26
>
> **Q1) FD is insufficient to assess the quality of interactions**
>
> Our evaluation primarily focused on human-human coordinated interactions, assessing how realistic the generated body poses and configurations are. However, to address explicit physical-plausibility evaluation, we define two additional metrics, *penetration ratio* and *contact distance*, and conduct further experiments.
>
> For human-human penetration, we evaluate each human mesh by measuring the ratio of its vertices that lie inside another human mesh. This ratio is then averaged over all individuals in the scene. Similarly, we evaluate each human mesh by measuring the ratio of its vertices that lie inside the object for human-object penetration.
>
> For the contact distance, we evaluate six categories: five hand-contact categories (board, box, bucket, chair, and desk) and one hip-contact category (bench). For each case, we randomly sample 10 vertices from the corresponding body region and compute their average distance to the object mesh. The final contact distance is obtained by averaging these per-case distances.
>
> Due to space constraints, please refer to the *Table 1, 2, 3* shown in the rebuttal for **Reviewer YRDq** for evaluation results on additional metrics.
>
> **Q2) Whether the collision loss penalizes correct interactions**
>
> *Table 1. The ratio (×1000) of interpenetrating vertices among SMPL-X humans. Lower values indicate less penetration.*
>
> | Method        | 3 Human | 4 Human | 5 Human |
> |---------------|--------:|--------:|--------:|
> | AABB-based (orignal version) |  15.68  |   **19.82**  |  **25.69**  |
> | Capsule-based |   **15.23**  |  25.66  |   28.72  |
>
> ---
>
> *Table 2. The ratio (×1000) of SMPL-X human vertices that penetrate the object. Lower values indicate less penetration.*
>
> | Method        | 3 Human | 4 Human | 5 Human |
> |---------------|--------:|--------:|--------:|
> | AABB-based (orignal version) |  **7.74**  |   **7.19**  |  10.18  |
> | Capsule-based |   7.77  |  8.41  |   **8.37**  |
>
> ---
>
> *Table 3. The contact distance (m) between a SMPL-X human and an object. Lower values indicate more precise contact.*
>
> | Method        | 3 Human | 4 Human | 5 Human |
> |---------------|--------:|--------:|--------:|
> | AABB-based (orignal version) |  0.031  |   **0.028**  |  **0.026**  |
> | Capsule-based |   **0.026**  |  0.028  |   0.030  |
>
> ---
>
> As mentioned in Lines 179-182, the collision loss is applied only between humans who are not explicitly specified to have an HHI (that is, only between those without "fine-grained interactions"), so its impact is minimal. However, to address reviewer’s concern and to eliminate even this minor influence, we test a new collision loss that approximates each human as a set of capsules and computes collisions accordingly. The process is as follows:
>
> Preliminary: The mathematical definition of a capsule is the set of all points equidistant from a line segment. Specifically, a capsule consists of a cylinder with radius $r$ around the central axis segment and two hemispheres with radius $r$ attached to the ends of the cylinder’s top and bottom circular faces.
>
> 1) We define the edges formed by the 21 SMPL-X joints as the central axis segments of each capsule. Additional edges are introduced to represent the hands and head. There are a total of 24 edges.
>
> 2) We optimize the radius $r_i$ for each capsule $C_i$. We use the same dataset (mentioned in Line 125) employed when training the pose encoder/decoder as ground truth, and use Chamfer Distance as the objective function.
>
> 3) Given $(\mathbf{R}, \mathbf{t}, \mathbf{s}, \theta)$, we can now compute the capsule-based human representation with minimal computation.
>
> 4) The collision between two capsules $C\_i$ and $C\_j$ is approximated as $\min(0, r\_i + r\_j - d\_{i,j})$, where $d\_{i,j}$ is the distance between their central axis segments.
>
> 5) The collision between two humans $\mathcal{H}\_1$ and $\mathcal{H}\_2$ is defined as the sum of pairwise capsule collisions between all capsules from each human.
>
> ---
>
> To compare our original AABB (axis‑aligned bounding‑box)–based collision loss with the newly proposed capsule‑based collision loss, we sampled HHOI scenes with each method and evaluated them in terms of penetration and contact quality.
> *Table 1* reports the average human-human penetration ratio, *Table 2* reports the average human-object penetration ratio, and *Table 3* reports the average human-object contact distance.
>
> The results show negligible differences between the two losses, indicating that, contrary to reviewer’s concern, the AABB‑based collision loss does not heavily penalize correct interactions.
>
> **Q3) Does the proposed dataset contribute to the improvement?**
>
> *Table 4. Comparison of our collected dataset.*
>
> | Dataset | # of categories | # of samples | Non-contact interaction objects |
> |---------------|--------:|--------:|--------:|
> | CORE-4D |   5  |  8160  |  X  |
> | Capture |   11  | 18860  |   O  |
> | Synthetic |   3  | 299  |   O  |
>
> ---
>
> First, to avoid confusion, we clarify that all samples belonging to each object category are collected from only one source — either CORE‑4D, our own capture system or synthetic generation.
>
> Because prior work on HHOI is scarce, publicly available datasets are almost nonexistent. CORE‑4D focuses mainly on scenarios where two people carry a single object together. However, we captured a new diverse dataset including scenarios in which HHOI occurs even without direct contact. Its significance lies not in simply boosting HHOI generation performance on models, but rather in enabling the modeling of HHOI scenarios that existing datasets cannot.
>
> *Table 4* compares the features of each collection pipeline. "# of categories" refers to how many categories of object are present in the dataset. "# of samples" refers to the total number of samples, including both HOI and HHI samples. "Non-contact interaction objects" refer to items such as "painting", which are involved in the scene context but do not require physical contact during the interaction. For objects that are difficult to capture in an indoor environment, we employed a synthetic data generation pipeline.
>
> In addition, to address the limited size of our current dataset, we plan to expand it by collecting more categories and samples in the future.
>
> **Q4) Does combining HOI and HHI to form HHOI require further training?**
>
> We emphasize that our advanced sampling enables sampling from the joint distribution of two different score-based models without additional training.
>
> **Q5) What datasets are used to train the HOI and HHI components respectively?**
>
> In brief, we train the HOI diffusion and HHI diffusion models using a combination of existing datasets (e.g., CORE-4D), data captured with our own system, and synthetic data generated using off-the-shelf models. Detailed information is provided in Sec. 4 of the main paper and Sec. A of the supplementary material.
>
> **Q6) Exact formulation of the variance loss**
>
> It follows the standard definition of variance: $\mathrm{Var}(\\{x\_i\\}\_{i=1}^N) = \sum\_{i=1}^N \frac{||x\_i - \mu||^2\_2}{N}, x\_i \in \mathbb{R}^n$.
>
> **Q7) How is the final result selected from $N + N(N–1)$ candidates?**
>
> Exactly $N+2M$ humans are sampled (where $M$ is the number of HHI pairs; see Lines 155-156 of the paper for the precise formulation). Since we mitigate the discrepancy between duplicate humans using the inconsistency loss, we finally take the average of the duplicate humans' $(\mathbf{R}, \mathbf{t}, \mathbf{s}, \theta)$.
>
> **Q8) How is the DAG constructed?**
>
> We first create a pickle file for HHOI sampling. The pickle generation program takes as input the number of humans, text prompts, and the HHI pairs (i.e., explicitly specifying that human $\mathcal{H}_i$ and $\mathcal{H}_j$ are involved in an HHI). It automatically verifies whether the specified HHI pairs form a valid DAG via a topological sort.
>
> **Q9) Why not use VPoser for pose embedding?**
>
> There are two main reasons why we trained a new body pose embedding network:
>
> 1) We want to keep the dimensionality of the body pose embedding low.
>
> 2) We aim to maintain a simple network architecture.
>
> In our experience, when the dimensionality of the body pose embedding exceeds 20D, HHOI diffusion shows a performance drop (VPoser uses a 32D embedding). Moreover, keeping the pose embedding network simple prevents a performance drop during the PF ODE process of our advanced HHOI sampling.
>
> **Q10) Why adopt a human scale instead of normalizing all objects to a unified scale?**
>
> Thank you for the valuable feedback! In fact, we consider handling humans of different scales as part of our future work, which is why we introduced the concept of human scale. However, regardless of whether human scale is used, reviewer’s suggestion to normalize object scales into a unified scale appears to be a promising approach for real-world applications. Since there is no fundamental difference, switching to that version would not be difficult and is worth trying.
>
> **Q11) Text-to-HOI and Text-to-HHI should not be claimed as contributions?**
>
> We emphasize that our contribution also lies in modeling HOI and HHI using a score-based diffusion approach, since these models serve as the components enabling coherent HHOI sampling. While previous methods have addressed these two tasks independently, merging the two individual models into a single model has been a challenge due to each model's unique structure.
>
> **Q12) What is the fundamental difference between HHOI and prior HHI tasks?**
>
> Compared to prior HHI tasks, HHOI explicitly incorporates objects, making the interaction scenarios more contextually concrete. For instance, a generic "two people talking" interaction can vary significantly in terms of interpersonal distance, body orientation, and poses, depending on whether they are sitting face-to-face at a table or side-by-side on a couch. By conditioning on the object context, our HHOI reduces such ambiguity.

---

> > ### Comment · Reviewer_w3tN · 2025-08-05
> >
> > Thank you for the detailed response. Most of my concerns have been addressed. However, there are still two minor issues.
> >
> > First, while I do acknowledge that integrating HOI and HHI into a unified system has its merits, I believe it is inappropriate to list the two individual modules separately as contributions of this paper, as stated in Line 46:
> > “(1) Text-to-HOI model that predicts a person’s relative position, orientation, body pose, and scale given an object and a text description; (2) Text-to-HHI model that predicts the interpersonal distance, relative orientation, and individual poses of two people.”
> > This could potentially mislead readers regarding the novelty of the work.
> >
> >
> > Additionally, is the constructed HHOI dataset expressed in metric units? Inconsistent units could affect the generalizability of the dataset.

---

> > > ### Author Response · Authors · 2025-08-05
> > >
> > > Thank you for the constructive feedback. We understand your concern and will revise our contribution statements accordingly. Regarding the unit system of our dataset, the HHOI dataset is expressed in metric units. We will revise both aspects in the final version to improve clarity.

---

### Official Review · Reviewer_YRDq · 2025-07-02

**Clarity:** 3
**Significance:** 2
**Originality:** 2
**Rating:** 4
**Confidence:** 4

**Summary:**

This work introduces a novel framework for generating Human-Human-Object Interactions (HHOIs), addressing the scarcity of diverse multi-person interaction data. The authors contribute a new 3D dataset captured with a multi-camera system, as well as a synthetic HHOI dataset generated using pretrained image diffusion models, especially targeting scenes that are difficult to capture in real life.

A score-based diffusion model unifies both real and synthetic data, enabling semantically grounded, realistic generation of dyadic and multi-person HOIs, conditioned on object categories and text descriptions. The approach is scalable to more complex social scenarios involving multiple people and shared objects.

**Questions:**

1. Why is the pose embedding dimension set to 126? So, where is the human root translation represented in the model?

**Ethical Concerns:**

["NO or VERY MINOR ethics concerns only"]

**Final Justification:**

Thank you to the authors for the detailed responses. Most of my concerns have been addressed, and I will raise my rating to BA.

**Limitations:**

yes

**Quality:**

2

**Strengths And Weaknesses:**

### **Strength**
1. Overall, the paper is well-structured and easy to follow.
2. The proposed method supports the generation of interactions involving 2 to 5 humans, and demonstrates good generalization capabilities to unseen scenes and textual descriptions.
3. Both the qualitative and quantitative results appear to outperform the baselines reasonably well.



### **Weakness**
1. While the visual results are promising on semantic-level, Figure 1 reveals noticeable interpenetration between humans, objects, and the ground.
2. There is no evaluation metric provided to quantify human-object interpenetration or contact accuracy, which are crucial for assessing the physical plausibility of the generated interactions.
3. Although the interaction categories in the CORE4D dataset are relatively limited, it contains dynamic interaction sequences, whereas the dataset used in this work focuses on static interactions. As a result, the two datasets differ in terms of difficulty and research contribution. It would be beneficial to include a comparative table that provides a statistical overview of the datasets, which could help clarify their respective scopes and highlight the novelty of the proposed dataset.

---

> ### Author Rebuttal · Authors · 2025-07-26
>
> *Table 1. The ratio (×1000) of interpenetrating vertices among SMPL-X humans. Lower values indicate less penetration.*
>
> | Method        | 2 Human | 3 Human | 4 Human | 5 Human |
> |---------------|--------:|--------:|--------:|--------:|
> | Ours          |   7.90  |   15.68  |   19.82  |  25.69  |
> | Ours+Refine |   **3.22**  |  **6.94**  |   **7.37**  |   **8.25**  |
> | GenZI |   13.91  |   24.60  |   21.55  |   49.06  |
> | Depth Opt. |   13.91  |   14.06  |   19.35  |   26.16  |
>
> ---
>
> *Table 2. The ratio (×1000) of SMPL-X human vertices that penetrate the object. Lower values indicate less penetration.*
>
> | Method        | 2 Human | 3 Human | 4 Human | 5 Human |
> |---------------|--------:|--------:|--------:|--------:|
> | Ours          |   5.49  |   7.74  |   7.19  |  10.18  |
> | Ours+Refine |   **2.78**  |  **2.77**  |  **3.67**  |   4.55  |
> | GenZI |   60.71  |   15.64  |   11.93  |   8.91  |
> | Depth Opt. |   11.48  |   4.15  |   9.41  |   **1.20**  |
>
> ---
>
> *Table 3. The contact distance (m) between a SMPL-X human and an object. Lower values indicate more precise contact.*
>
> | Method        | 2 Human | 3 Human | 4 Human | 5 Human |
> |---------------|--------:|--------:|--------:|--------:|
> | Ours          |   **0.029**  |   **0.031**  |   **0.028**  |  **0.026**  |
> | Ours+Refine |   0.032  |   0.038  |   0.041  |   0.038  |
> | GenZI |   0.103  |   0.126  |   0.537  |   0.389  |
> | Depth Opt. |   0.666  |   1.992  |   4.415  |   2.736  |
>
> ---
>
> **Q1) Evaluation for physical plausibility**
>
> Our evaluation primarily focuses on human-human coordinated interactions, assessing how realistic the generated body poses and configurations are. However, to address concerns about the absence of explicit physical-plausibility evaluation, we define two additional metrics, *penetration ratio* and *contact distance*, and conduct further experiments.
>
> For human-human penetration, we evaluate each human mesh by measuring the ratio of its vertices that lie inside another human mesh. This ratio is then averaged over all individuals in the scene. Similarly, we evaluate each human mesh by measuring the ratio of its vertices that lie inside the object for human-object penetration. *Table 1* reports the human-human penetration ratio, and *Table 2* reports the human-object penetration ratio.
>
> For the contact distance, we evaluate six categories: five hand-contact categories (board, box, bucket, chair, and desk) and one hip-contact category (bench). For each case, we randomly sample 10 vertices from the corresponding body region and compute their average distance to the object mesh. The final contact distance is obtained by averaging these per-case distances. *Table 3* reports the contact distance.
>
> We randomly sampled 10 scenes per (category, human-count) combination.
>
> *Table 1, 2, 3* show that our model outperforms competitors in terms of penetration and contact even before refinement (cover this in **Q2**). Although the penetration ratio for Depth Optimization looks favorable when sampling three or more humans, this is an illusion: Depth Optimization often places humans far from the object, thereby lowering the penetration ratio. This becomes clear in the contact distance results in *Table 3*. Especially, for the 5 human case, Depth Optimization’s contact distance is about 100 times larger than that of our method.
>
> **Q2) Human-Ground penetration**
>
> We would like to note that the surrounding environment, including the ground, is not used as input to our model, and it is used as decorative content for rendering. We'll revise this to avoid confusion.
>
> **Q3) Human-Object penetration**
>
> Regarding human-object penetration, the additional terms $\mathcal{L}\_\text{inc}$ and $\mathcal{L}\_\text{col}$ in our advanced PF ODE procedure (Eq. (16) in the main paper) may slightly hinder the recovery of the original HOI and HHI distributions. To address this, we introduce an additional simple post-processing step after HHOI sampling that optimizes translations to reduce inter-mesh penetration. The optimization objective is as follows:
>
> $\min\_{\\{t_i\\}\_{i=1}^N} \mathcal{L}\_\text{pen}(\\{t_i\\}\_{i=1}^N) + \lambda\sum\_{i=1}^N ||t\_i||\_2^2$,
>
> where $t\_i$ is translation vector of human $i$, $\mathcal{L}\_\text{pen}(\\{t_i\\}\_{i=1}^N)$ is total penetration between all meshes, and $\lambda$ is weight factor for regularization term.
>
> As shown in *Tables 1* and *Table 2*, the post-processing noticeably reduces penetration. The contact distance in *Table 3* increases slightly, but the change is very small and therefore negligible.
>
> **Q4) Dataset comparison & statistics**
>
> *Table 4. Comparison of CORE-4D and our own collected dataset.*
>
> | Dataset       | Frame type | # of categories | Non-contact interaction objects |
> |---------------|--------:|--------:|--------:|
> | CORE-4D [36] |   Sequence  |  6  |  X  |
> | Capture (ours) |   Single  | 11  |   O  |
> | Synthetic (ours) |   Single  | 3  |   O  |
>
> ---
>
> *Table 4* compares the features of our newly collected data with those of CORE-4D. "Frame type" refers to whether the dataset is composed of single frame or sequential motion. "# of categories" refers to how many categories of object are present in the dataset. "Non-contact interaction objects" refer to items such as "painting", which are involved in the scene context but do not require physical contact during the interaction.
>
> We captured our own dataset primarily due to the limited object categories in CORE-4D, which includes only a few manipulative objects such as "box" or "board". In contrast, our captured dataset incorporates a wider variety of non-contact objects such as "campfire" and "whiteboard". For objects that are difficult to capture in an indoor environment, we employed a synthetic data generation pipeline.
>
> **Q5) Regarding human pose representation**
>
> The pose embedding is 10D, not 126D (see Line 126). The 126D refers to the original dimensionality of the body pose, which comes from the 6D rotation representation of 21 body joints (see Line 121). The root orientation and translation of human $\mathcal{H}$ are represented by $\mathbf{R}\_{\mathcal{H}}$ and $\mathbf{t}\_\mathcal{H}$ as in Eq. (1) of the main paper.

---

> > ### Comment · Reviewer_YRDq · 2025-08-04
> > **Comment by Reviewer YRDq**
> >
> > Thank you to the authors for the detailed responses. Most of my concerns have been addressed, and I will raise my rating to BA.

---

### Official Review · Reviewer_KKEh · 2025-07-03

**Clarity:** 2
**Significance:** 2
**Originality:** 3
**Rating:** 4
**Confidence:** 4

**Summary:**

The paper primarily focuses on generating interactions between multiple humans and a single object, implemented within a simulator. The core framework relies on score-based diffusion models and ODE sampling.
The approach first processes human-object interactions (HOI) and human-human interactions (HHI) separately, then integrates them into a unified framework for generating human-human-object interactions (HHOI). The model design incorporates considerations for datasets, collision avoidance, and interactive dynamics.

**Questions:**

In Figure 1, human feet are embedded in the floor. Is this because the scene is not used as input information? What role does the complete scene play besides decoration?

In Figure 3, humans and wheelchairs show model penetration. Does this indicate issues with the initial HOI module or the integrated HHOI optimization? The paper mentions collision handling only for human-human interactions, suggesting a stronger focus on human-human rather than human-object interactions.

Line 279 claims high-quality results in dynamic scenes, but according to the description in Figure 4, HHOI only generates single final frames, with continuous frames processed by other methods, failing to demonstrate the dynamic capability of this method. It seems that objects are merely treated as fixed elements-—what significant differences does this approach have from ordinary human-scene generation methods like Narrator (ICCV 2023)?

The paper mentions object inputs ambiguously: "object category" in lines 40-41, "object instance mesh" in line 100, and "object context" in line 304. Is the model trained for each "object category", with "object instance mesh" input during testing? The pipeline should clarify input/output definitions.

**Ethical Concerns:**

["NO or VERY MINOR ethics concerns only"]

**Final Justification:**

The authors have addressed most of my previous concerns. Hence, I will keep my rating.

**Limitations:**

yes

**Quality:**

3

**Strengths And Weaknesses:**

Strengths:
The authors have expanded upon HOI and HHI, demonstrating in-depth thinking on the HHOI problem by considering factors such as indirect implicit interactions, physical collision constraints, inconsistency constraints, and motion priors. The dataset design is comprehensive, incorporating two types: multi-view laboratory HHOI data and diversely generated HOI/HHI data from images. The experimental design is rational, and the results align with expectations.

Weaknesses:
This paper appears to overstate its claims, particularly regarding high-quality results in dynamic scenes. While the proposed HHOI method generates single final frames, the processing of continuous frames relies on auxiliary techniques, raising concerns about its dynamic capability. Additionally, the treatment of objects as static elements rather than dynamically interacting entities further limits the perceived innovation. To strengthen this paper, it would be valuable to explicitly highlight the key distinctions between this approach and prior human-scene generation methods, such as Narrator (ICCV 2023), particularly in terms of temporal coherence and object-scene interaction.

---

> ### Author Rebuttal · Authors · 2025-07-27
>
> **Q1) Clarification on the use of the term "dynamic"**
>
> Regarding the use of the term "dynamic" in Line 279, we intended it to refer to single-frame scenes where the object is actively being used and manipulated by humans, and is thus in direct contact with them. Line 270 describes our original intention. However, since "dynamic" is often interpreted as motion across continuous frames, we will revise the wording in the final version to better reflect our intended meaning.
>
> Regarding the motion generation component, we clarify that the result shown is conditioned on our model's output and produced by utilizing Diffusion-Noise-Optimization (CVPR 2024). This is presented as a potential downstream application rather than a core contribution of our paper. We will update the summary of contributions accordingly to avoid misunderstanding.
>
> **Q2) Differences between our work and Narrator (ICCV 2023)**
>
> Narrator generates a single frame of human-scene interaction, taking a text description and a scene mesh as input. It is trained on the single-person human-scene interaction dataset PROX (ICCV 2019), but can produce multi-person scenes via collision-based optimization.
>
> Our model produces a similar type of output, that is a single frame of multiple people interacting in the presence of an object. However, the key difference lies in the nature of the interactions: specifically, our model captures human-human interactions, while Narrator does not.
>
> Because Narrator is trained on a single-person dataset, it can place multiple people in the scene with appropriate spacing, but it lacks the ability to generate meaningful interactions between them. For example, it may generate two people standing in front of a whiteboard, but it cannot depict them actively engaging with each other (e.g., talking, gesturing, or making eye contact). In contrast, our model can synthesize plausible configurations of body pose, orientation, and interpersonal distance that correspond to genuine human-human interactions, which is difficult to achieve when training only on single-person pose data.
>
> **Q3) Human-Object penetration**
>
> *Table 1. The ratio (×1000) of SMPL-X human vertices that penetrate the object. Higher values indicate more penetration.*
>
> | Method          | 1 Human | 2 Human | 3 Human | 4 Human | 5 Human |
> |-----------------|---------|---------|---------|---------|---------|
> | HOI             | 5.39 | -       | -       | -       | -       |
> | HHOI            | -       | 5.49 | 7.74 | 7.19 | 10.18 |
> | HHOI (Refine)   | -       | **2.78** | **2.77** | **3.67** | **4.55** |
>
> ---
>
> We perform additional experiments to analyze the cause of the human-object penetration.
>
> *Table 1* reports a quantitative evaluation of human-object penetration: for each (method, human count) combination, we randomly selected 160 scenes (16 categories × 10 scenes each) and measured the fraction of human mesh vertices that penetrate the object (scaled by ×1000). The penetration ratio for each scene is measured as the average of each human-object penetration in the scene.
>
> As shown in *Table 1*, the penetration ratio increases as the number of humans sampled in HHOI sampling grows. This is because, in our advanced PF ODE procedure (Eq. (16) in the main paper), additional terms such as $\mathcal{L}\_\text{inc}$ and $\mathcal{L}\_\text{col}$ can slightly hinder the recovery of the original HOI and HHI distributions.
>
> To address this, we introduce an additional post-processing step after sampling that optimizes translations to reduce inter-mesh penetration. The optimization objective is as follows:
>
> $\min\_{\\{t_i\\}\_{i=1}^N} \mathcal{L}\_\text{pen}(\\{t_i\\}\_{i=1}^N) + \lambda\sum\_{i=1}^N ||t\_i||\_2^2$,
>
> where $t\_i$ is translation vector of human $i$, $\mathcal{L}\_\text{pen}(\\{t_i\\}\_{i=1}^N)$ is total penetration between all meshes, and $\lambda$ is weight factor for regularization term.
>
> The last row of *Table 1* shows that the human-object penetration is greatly improved after this optimization.
>
> **Q4) Role of the background scenes**
>
> Our model considers the relative configuration to the target object, and other parts of the scene, including the floor, are not considered. The surrounding environment is shown purely for aesthetic purposes to provide contextual understanding of the scene.
>
> **Q5) Clarification regarding the role of objects in the model input/output**
>
> Our model’s input is text description only, and object meshes are not provided as input. However, because the training data (both HOI data and HHI data) is organized per specific mesh instance, we wrote “object instance mesh is provided” on Line 101. We will revise the inputs/outputs description for clarity.

---

> > ### Comment · Reviewer_KKEh · 2025-08-04
> > **Comment by Reviewer**
> >
> > Thanks for the rebuttal. The authors have addressed most of my previous concerns. Hence, I will keep my rating.

---

### Comment · Area_Chair_wRwc · 2025-08-05
**Reminder: Please follow up on the authors’ rebuttal**

Dear Reviewers,

Many thanks to Reviewers KKEh, YRDq, and hSov for the thoughtful discussion so far.

Reviewer w3tN, just a gentle reminder – we’d greatly appreciate it if you could take a moment to review the authors’ responses and share any follow-up thoughts.

With the discussion phase wrapping up soon, your timely input can help ensure a productive exchange and give the authors a final chance to clarify any remaining points.
Thanks again for your time and contributions to the NeurIPS review process.

Best,

Your AC

---

### Note · Authors · 2025-08-13

We sincerely thank all the reviewers for their constructive feedback and insightful discussions, and we also appreciate the Area Chair for facilitating a seamless discussion process.

The reviewers recognized the novelty of our HHOI formulation: "expanded upon HOI and HHI, demonstrating in-depth thinking on the HHOI problem" (R1=KKeh), "proposal of HHOI generation is novel" (R4=hSov). Our methodology was also considered meritable by the reviewers: "Score-based framework merges HOI and HHI to generate coherent HHOI" (R2=YRDq, R3=w3tN, R4) Reviewers also acknowledged the value of our collected dataset: "proposed dataset may be valuable for future research on HHOI" (R3), "The dataset design is comprehensive, incorporating two types" (R1).

All reviewers mentioned that their concerns on the paper have been addressed accordingly in the rebuttal. The modifications suggested in the reviews will be incorporated in the final revision. The code and dataset will be released prior to the conference.

---

### Decision · Program_Chairs · 2025-09-17

**Decision:**

Accept (poster)

**Comment:**

The reviewers agreed that the problem addressed in the paper is meaningful, the proposed method and dataset are valuable, and the results are encouraging. The rebuttal satisfactorily resolved most of the reviewers’ concerns, and acceptance was recommended by all. The authors are encouraged to enhance the final version of the paper by taking into account the reviewers’ suggestions.